# Frequency Hopping Signals Tracking and Sorting Based on Dynamic Programming Modulated Wideband Converters

**Ziwei Lei \*, Peng Yang, Linhua Zheng, Hui Xiong and Hong Ding**

College of Electronic Science, National University of Defense Technology, Changsha 410073, China

\* Correspondence: leiziwei08@nudt.edu.cn; Tel.: +86-731-8700-3273

**Abstract:** Most of the earlier tracking and network sorting approaches with a high sampling rate for frequency hopping (FH) signals did not adapt to the wideband system during their implementation, whereas the sub-Nyquist based algorithms cannot satisfy the real-time requirement for dealing with the rapid change of sparsity. It is important to improve the compressed sensing (CS) methods for tracking and sorting wideband FH signals. In this paper, a dynamic programming modulated wideband converters (MWC) scheme is proposed. First, considering the wide gap of FH signals, an improved power estimation method is proposed to track the support set in the time domain. Second, to sort multiple signals more effectively, a feedback control algorithm based on dynamic programming is proposed. In the proposed method, the total sampling rate is decreased significantly, and multiple FH signals are separated rapidly without recovery based on the results of tracking and comparative power. Simulations show that the proposed method can track and sort FH signals efficiently and more practically than previous methods.

**Keywords:** frequency hopping signals; dynamic programming; modulated wideband converters; tracking; sorting

---

## 1. Introduction

Frequency hopping (FH) communication is one of the main types of spread spectrum communication [1]. The carrier frequency of the FH signal hops under the control of a pseudo-noise (PN) code, by which the spectrum is spread. FH communication systems have been widely used in military communications and covert communications due to their strong anti-interference ability and low probability of interception. Tracking and network sorting of multiple FH signals are important tasks to obtain useful information in communication reconnaissance. The tracking of FH signals refers to rapid estimation of the carrier frequency of the signal when the signal is hopping [2]. There are two aspects of FH signal tracking: detecting the timing of carrier frequency hopping, which means estimating the hop timing and estimating the frequency of the new hop. Thus, tracking of FH signals can be regarded as a real-time estimation of hop timing and frequency. The purpose of the network sorting is to separate multiple FH signals into their own sources [3]. Network sorting depends on estimating the unique parameters of each signal and the independence of signals. Thus, it is a problem of information processing based on the characteristics of signals. A number of blind signal tracking and sorting structures have been proposed for FH signals. One widely used method is spatial time-frequency analysis, which is usually utilized to describe sparse signals in the time-frequency domain [4]. In [5], the sparsity of the FH signal in the time-frequency domain was described and the parameters of multiple frequency hopping signals were estimated via sparse linear regression. In [6–10], the time of arrival (TOA), direction of arrival (DOA), frequency and other parameters were

estimated based on time-frequency analysis. In [6,7], joint signal parameter estimation of FH signals was proposed. In [8], a blind source separation approach exploiting the differences in time-frequency signatures of the sources to be separated was proposed, which is based on the diagonalization of a combined set of spatial time-frequency distributions and is applicable to FH signals. In [9], a kind of blind separation algorithm of FHSS signals with a time-frequency ratio of the matrix based on short-time Fourier transform (STFT) was proposed, which calculated the elements of the mixing matrix in the frequency domain and gave the solution of the inverse matrix directly. In [10], the proposed method uses the smoothed pseudo Wigner–Ville distribution (SPWVD) of each separated FHSS signal to estimate its transmission parameters and separates the signals by the joint approximate diagonalization of eigen-matrices (JADE) algorithm. In [4], a novel method to detect single source points in the time-frequency domain for separation was proposed, which also improves the subspace projection method to recover signal [4,6–10]. These methods are intuitive and effective. It can be concluded that estimating parameters is a way to obtain the conditions for tracking and sorting. However, the methods in [4,6–10] cannot process signals in real time, because they are all based on clustering information of a large amount of data. In many practical applications, such as jamming and dynamic reconnaissance, these methods do not satisfy the command. For FH signal tracking, Liu and Wang proposed a tracking model for FH signals, by which the frequency of multiple FH signals can be tracked in real time based on the temporal autoregressive moving average (ARMA) and sparse Bayesian learning (SBL) model [2,11]. However, they suffer from enormous computing in practice. As a result, the computational complexity may exceed the capability of the hardware. To solve these problems, one method is to track and separate the signals depending on their irrelevance and another is to reduce the sampling rate of processing.

As the network sorting of FH signals is a special case of the blind source separation (BSS) problem, some universal approaches for BSS are effective. For instance, independent component analysis (ICA) has better results on stationary and independent signals [12–14]. In [12], the ICA algorithm was introduced, which can be considered to be a foundation for further separation methods. In [13,14], a FastICA algorithm was introduced for separation, which is more efficient than the traditional ICA. The correlation coefficient was taken as the evaluation criterion of the separation performance. Although these separation methods based on ICA do not need to obtain the array structure beforehand, they constrain the number of sources to be less than the number of sensors. In many practical applications, this condition is not satisfied. Furthermore, these methods cannot take advantage of sparsity in the processing and the signal-to-noise ratio (SNR) is poor. Recently, tensor decomposition [15], subspace-based projection [3] and neural network [16] were used for sorting FH signals. However, all of these methods start from the premise that the total sampling rate of channels or array receivers should meet or exceed the Nyquist sampling rate. The sampling rates of analog-to-digital converters (ADCs) can be relatively high to meet the Nyquist theorem. Meanwhile, large amounts of data must be cached for calculation and the amount of computations for further digital processing is enormous.

Compressed sensing (CS) has been applied for wideband sparse signal processing to overcome the obstacle of high sampling rate [17]. Using CS to separate signals that are sparse in the time-frequency domain, such as radar, linear frequency modulation (LFM) and FH signals, is a hot topic in recent research [18–22]. The core idea of these methods is to recover the time-frequency representation of signals. Thus, the sparse signals can be separated in the time-frequency domain. However, these methods cannot process the signals in real time and certain time-frequency information is required in advance, because the signals must be recovered before further processing and it takes a lot of computation. It can be concluded that CS has great advantages in practical applications, but studies on the tracking and separation of FH signals using CS are scarce and not well-rounded. The major difficulties in using the classical CS theory for the tracking and sorting of FH signals are as follows:

1.  The traditional methods are not suitable for FH signals due to sparsity.

2. The structure of the traditional sensing matrix is planned and cannot be configured according to the signals.

3. The CS structures change the phase and amplitude characteristics of the signals. As a result, the performance of the method based on these parameters cannot be guaranteed.

4. The traditional CS structures require caching data for recovery, which is not suitable for real-time signal processing.

Nowadays, the problem of tracking time-varying sparse signals based on sparse signal processing and dynamic CS has been tackled. Dynamic CS has been applied in dynamic magnetic resonance imaging (MRI) [23], underwater channel estimation [24], and radar imaging [25]. In [26,27], tracking and dynamic filtering of time-varying sparse signals are analyzed under certain assumptions. In [28,29], a series of universal approaches were proposed for online optimization in dynamic environments. These methods transform the CS of time-varying sparse signals to online convex optimization, which proceed from the perspective of regret analysis. For FH signals, a time-frequency analysis based on sparse recovery was proposed and the unconstrained sparse representation model of FH signals was established according to the punish function theory [30]. However, these dynamic CS methods have difficulties processing FH signals. First, since the carrier frequency of FH signals is under the control of a PN code, the iterations of solving the optimization problem will increase under the condition of full blindness, which will affect the real-time tracking. Secondly, the punish function cannot adapt to the changes in signal sparsity.

Modulated wideband converters (MWC) structure is a sub-Nyquist sampling scheme for acquiring sparse wideband signals [31,32]. It is flexible and can be realized by commercial components. Many improved structures, such as distributed MWC (DMWC) [33], random partial Fourier structured MWC (RPFMWC) [34], and array based MWC [35], have been proposed for specific signal processing. In our previous work [36], a method for detecting FH signals based on MWC was proposed, based on channelization and energy detection. However, the method focuses on single-target FH signal detection and is not applicable to multi-target signals. Furthermore, the method mainly processes the signals in the frequency domain and losses the time information. In this paper, we propose methods for the tracking and network sorting of FH signals based on the estimation of TOA and power by MWC in the time domain. A dynamic programming multi-frequency function is designed and applied in the system to identify the signals and obtain the support set directly. Instead of caching data for further operations, the proposed system processes the input signals in real time, which is necessary for signal tracking. Furthermore, separation of the multiple signal is implemented by the results of tracking and comparative power.

The rest of this paper is organized as follows. In Section 2, we provide the basic information related to multiple FH signals. A detailed design of the dynamic programming MWC method for signals tracking and sorting is presented in Section 3. The effects of some parameter settings and simulation results are given in Section 4 and the conclusions are presented in Section 5.

## 2. Problem Formulation

Assume that there are $R$ FH signals impinging on a single antenna, the $r$th FH signal can be expressed as

$$x_r(t) = \alpha_r(t)e^{j(2\pi f_r(t)t + \phi_r(t))}, \quad 0 < t \leq T \tag{1}$$

where $\alpha_r(t)$ is the complex envelope, $f_r(t)$ is the FH instantaneous frequency, $\phi_r(t)$ is the phase of the signal, and $T$ is the observation time. The received multiple FH signals by a single channel can be expressed as

$$x(t) = \sum_{r=1}^{R} x_r(t) + n_r(t), \quad 0 < t \leq T \tag{2}$$

where $n_r(t)$ is additive white Gaussian noise. According to actual situations, FH signals are assumed to be relatively power stable, which means that all the hops of the same source have similar power, and the signals do not collide.

As shown in Figure 1, the task of the algorithm is to track and separate FH signals under sub-Nyquist sampling. The tracking results and comparative power are the basis of the separation.

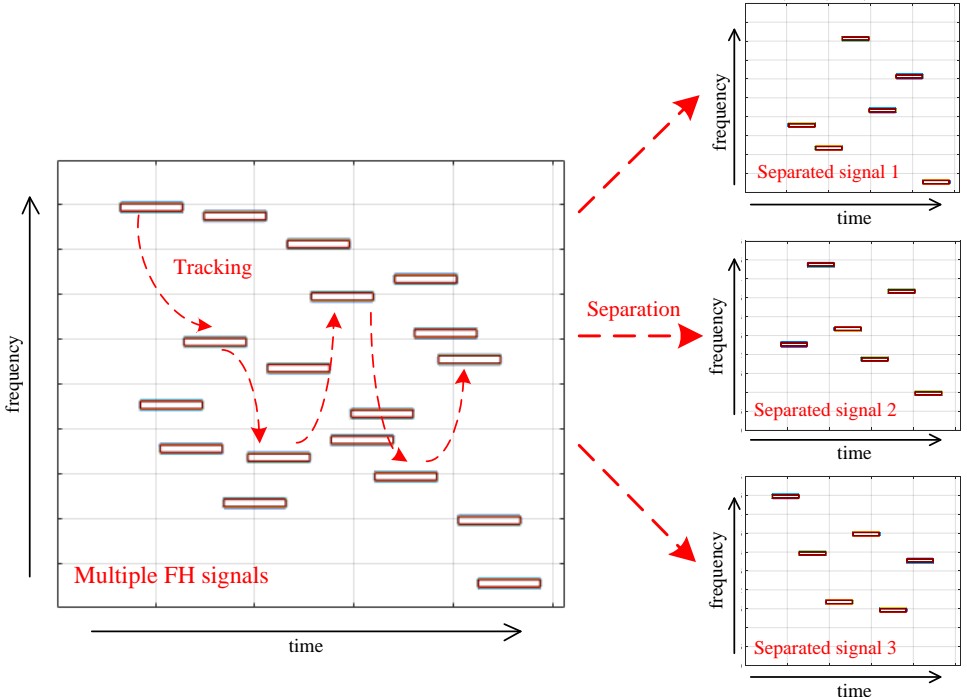

**Figure 1.** Sketch of tracking and separation of frequency hopping (FH) signal.

## 3. Structure Design for FH Signal Tracking and Sorting

In this section, first, an improved MWC structure is applied to track sub-channels with signals in the time domain, and second, a dynamic programming algorithm is proposed for the sorting.

### 3.1. Tracking of FH Signals

Tracking of signals is a fundamental requirement for a real-time system. The purpose of FH signal tracking is to obtain the time and frequency of new hops as soon as possible, which is an important basis for separation. In order to apply CS to tracking, the system should be able to cope with changes in sparsity and update the processing results rapidly. Although the classic MWC structure works in the Fourier domain and is fit to conduct wideband spectrum sensing, the under-sampling board is designed by a given sparsity [33]. To make the system more suitable and feasible for FH signals, a novel dynamic programming MWC structure is proposed, which is depicted in Figure 2.

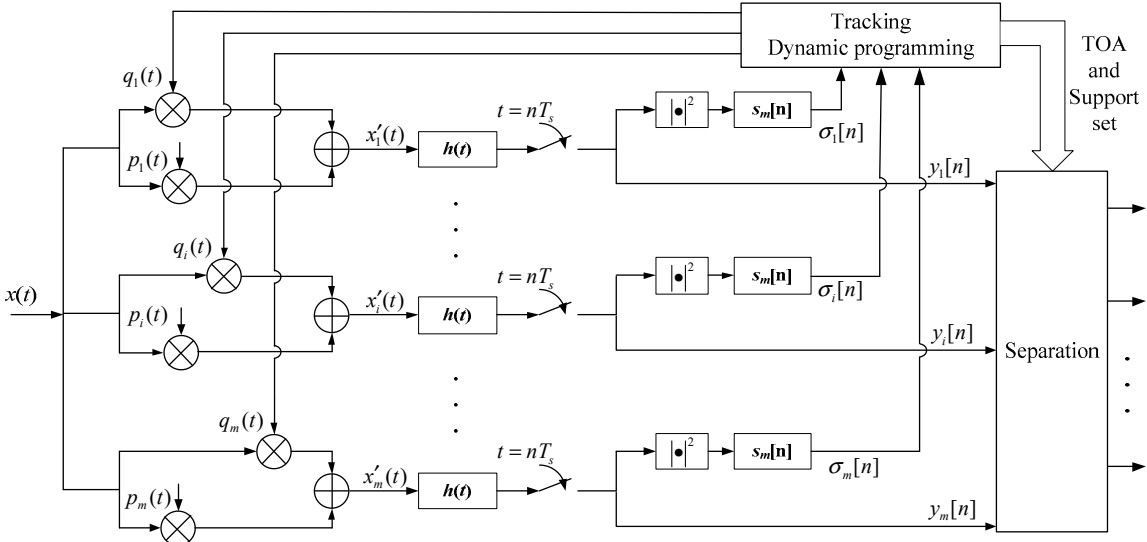

**Figure 2.** Dynamic programming modulated wideband converter (MWC) structure.

As shown in Figure 2, the dynamic programming MWC structure is a single antenna system consisting of $m$ parallel sub-channels. In each sub-channel, the multiband signal $x(t)$ is multiplied by a different mixing function $p_i(t)$ and multi-frequency function $q_i(t)$; $h(t)$ is an ideal low-pass filter and $s_m[n]$ is a smoothing filter that performs real-time sliding accumulation of energy. Once the mixing function of the sub-channel is determined, the response of the channel is relatively determined. Because of the pseudo-random property and expectation restricted isometry property (ExRIP) of the mixing function, $p_i(t)$ is restricted and has certain random features [37]. The precise description of $p_i(t)$ is provided in Appendix A and a brief description of ExRIP is provided in Appendix B. It is difficult to distinguish the signal information only by the mixing function. To solve these problems, a multi-frequency function $q_i(t)$ is designed and implemented that depends on the mixing function and communication environment and is dynamically programmed by the estimation results of $\sigma_i[n]$. Unlike the randomness of the mixing function, the multi-frequency function is determined for judgment in each sub-channel and strengthens the sensing ability for certain sub-bands in different sub-channels. As FH signals have wide frequency gaps, the frequency hopping can be sensed by tracking the power change of the sub-band. Thus, with a reasonable distribution of the sub-bands to the sub-channel, the frequency hopping can also be tracked by the power estimation of sub-channels.

The discrete time Fourier transform (DTFT) of $y_i[n]$ can be expressed as

$$Y_i\left(e^{j2\pi f T_s}\right) = \sum_{l=-L_0}^{+L_0} c_{il} X(f - l f_p) + \sum_{l=-L_0}^{+L_0} d_{il} X(f - l f_p) \quad f \in \mathcal{F}_s \tag{3}$$

where $c_{il}$ and $d_{il}$ are the Fourier series coefficients of $p_i(t)$ and $q_i(t)$, respectively, and $d_{il}$ can be expressed as

$$d_{il} = \frac{1}{T_p} \int_0^{T_p} q_i(t) e^{-j2\pi l f_p t} dt \tag{4}$$

It is clear that $q_i(t)$ is a $T_p$–periodic function. In practical application, $q_i(t)$ can be simply set as

$$q_i(t) = \sum_l \lambda_{il} e^{j2\pi l f_p t} \quad k\frac{T_p}{M} \le t \le (k+1)\frac{T_p}{M} \tag{5}$$

where $M$ is the number of equal time intervals and $\lambda_{il}$ is the amplitude of $q_i(t)$. For each sub-channel, $\lambda_{il}$ is dynamically programmed for estimation of the support set, which will be discussed in Section 3.2. The algorithm of signal tracking in the time domain is described as follows:

Suppose there is no frequency hopping during the observation. Thus, the mixing signal $x(t)$ is a wide-sense stationary process (WSS), the Fourier transform of which is $X(f)$. $x_i(t)$ is a part of $x(t)$ and $X_i(f)$ is the Fourier transform of $x_i(t)$. The power spectrum of $x(t)$ can be expressed as:

$$P_x(f) = E\left[|X(f)|^2\right] = E\left[|X_i(f)|^2\right] \tag{6}$$

where $E[\bullet]$ denotes expectation and $P_x(f)$ is the Fourier transform of the autocorrelation function:

$$P_x(f) = \int_{-\infty}^{+\infty} r_x(\tau)e^{-j2\pi f\tau}d\tau \tag{7}$$

Based on [38], the power spectrum of the WSS signal is only related to itself and the result of cross correlation is equal to the power spectrum:

$$E\left[X_i(f_1)X_i^*(f_2)\right] = P_x(f_1)\delta(f_1 - f_2) \tag{8}$$

where $(\bullet)^*$ denotes conjugate and $f_1$, $f_2$ are arbitrary frequencies in the band of $x(t)$.

Thus, the power spectrum of $y_i[n]$ can be expressed as:

$$P_{y_i}(f) = E\left[|Y_i(f)|^2\right] = \sum_{l=-L_0}^{+L_0} |c_{il} + d_{il}|^2 P_x(f - lf_p) \quad f \in \mathcal{F}_s \tag{9}$$

Based on Parseval theorem,

$$\sum_{n=-\infty}^{+\infty} |y_i[n]|^2 = \frac{1}{2\pi} \int_{-\infty}^{+\infty} |Y_i(f)|^2 df \tag{10}$$

As $h(t)$ is an ideal low-pass filter, all the power of $y_i[n]$ is limited in $\mathcal{F}_p = \left[-\frac{f_p}{2}, \frac{f_p}{2}\right]$. Thus:

$$\begin{aligned}
E\left[\sum_{n=-\infty}^{+\infty} |y_i[n]|^2\right] &= E\left[\frac{1}{2\pi} \int_{-\infty}^{+\infty} |Y_i(f)|^2 df\right] \\
&= \frac{1}{2\pi} \int_{-f_p/2}^{f_p/2} E\left[|Y_i(f)|^2\right] df \\
&= \frac{1}{2\pi} \int_{-f_p/2}^{f_p/2} P_{y_i}(f) df
\end{aligned} \tag{11}$$

Thus, the estimation of signal power in the time domain and of the MWC sub-channel in the frequency domain are connected. In the proposed structure, we obtain

$$\sigma_i[n] = |y_i[n]|^2 \otimes s_m[n] = \sum_{n=n-\lceil l_s/2\rceil+1}^{n+\lceil l_s/2\rceil} a_s|y_i[n]|^2 \tag{12}$$

where $l_s$ is the length of the smoothing filter $s_m[n]$ and $a_s$ is the amplitude, and $\otimes$ denotes convolution. Combining Equations (11) and (12), it can be concluded that $\sigma_i[n]$ is a power estimation of $y_i[n]$. For an appropriate filter length $l_s$, $\{\sigma_i[n]\}$ $i = 1, 2, \ldots, m$ can track the power change between sub-channels. Based on Equation (9), the power of certain sub-bands can be reflected in $\{\sigma_i[n]\}$ $i = 1, 2, \ldots, m$. Therefore, as long as the signal-to-noise ratio (SNR) is high enough in its sub-band, the frequency

hopping of the signal can be detected and tracked by estimating the energy of the sub-channels in the time domain.

Suppose the threshold of energy detecting is $\varepsilon$, which is based on the length of the smoothing filter and original signal power. The TOA estimation of a single hop is

$$T_H[n] = \frac{\frac{(n-1)+n}{2} + \frac{(n+\tau-1)+n+\tau}{2}}{2} = \frac{2n+\tau-1}{2}$$
$$if \begin{cases} \sigma_i[n-1] > \varepsilon, & \sigma_i[n] \le \varepsilon \\ \sigma_j[n+\tau-1] \le \varepsilon, & \sigma_j[n+\tau] > \varepsilon \end{cases} \tag{13}$$

where $i, j$ $(i \neq j)$ are the sub-channel numbers, $\tau$ is a positive integer and less than the length of the smoothing filter.

Suppose there is a single hop $s_i(t)$ in sub-band $l_a$ and there are $\{l_k\}$ sub-bands including $l_a$, amplified in sub-channel $m_c$. Set $\lambda_{m_c l_i} = 0$ $l_i \notin \{l_k\}$ . Thus, the expectation of the output SNR $\eta_{oc}$ of $\sigma_{m_c}[n]$ is

$$\eta_{oc} = \frac{\left(d_{m_c l_a} + c_{m_c l_a}\right)^2}{card(\{l_k\}) \bullet (d_{m_c} + c_{m_c})^2 + (L - crad(\{l_k\})) c_i^2} \eta_i \tag{14}$$

where $card(\bullet)$ denotes the cardinality of sets, $d_{m_c}$ and $c_{m_c}$ are the expectation of the Fourier series coefficients of $p_{m_c}(t)$ and $q_{m_c}(t)$, respectively, in sub-channel $m_c$, and $\eta_i$ is the expectation of input equivalent narrowband SNR of the signal. However, if we bypass the multi-frequency function, we obtain

$$\eta_o = \frac{c_i^2}{L c_i^2} \eta_i = \frac{1}{L} \tag{15}$$

where $c_i$ is the expectation of the Fourier coefficient of $p_i(t)$. If $d_i \gg c_i$, the output SNR of sub-channel $m_c$ could be significantly increased.

It is more efficient to process directly in the time domain than in the frequency domain, which is conducive for signal tracking. At the same time, FH signals generally have a high equivalent narrowband SNR, which meets the requirement of processing.

### 3.2. Sorting of FH Signals

Based on the tracking algorithm in Section 3.1, the estimated hopping time is obtained. However, the power estimation of $\sigma_i[n]$ is not an estimation of source because the received signal $x(t)$ is modulated by $p_i(t)$ and $q_i(t)$. Furthermore, the multiple signal is not separated at the output. To solve these problems, in this section, the dynamic programming method is taken for the support set estimation of each single hop. The signals are separated according to their own sources based on hopping time and source power.

Equation (3) can be given by the following matrix form [31]

$$\mathbf{Y}(f) = \mathbf{A}\mathbf{Z}(f) \quad f \in \mathcal{F}_s \tag{16}$$

where $\mathbf{Z}(f) = [X(f - L_0 f_p), \cdots, X(f), \cdots, X(f + L_0 f_p)]^T$ and $\mathbf{A}$ is the sensing matrix. It is clear that $X(f + l f_p), l = -L_0, \ldots, 0, \ldots, L_0$ are independent and arranged in order of sub-bands. The support set can be regarded as the active sub-bands that contain the signals. The algorithms for solving the support set are the core of separation.

Suppose the support set is $S$, Equation (16) can be transformed to [31]

$$\mathbf{Y}(f) = \mathbf{A}_S \mathbf{Z}_S(f) \quad f \in \mathcal{F}_s \tag{17}$$

As the source signal is sparse, the number of elements in $S$ is smaller than the number of observed signals. Thus [31]

$$\mathbf{z}_s[n] = \mathbf{A}_S^\dagger \mathbf{y}[n]$$
$$z_l[n] = 0, l \notin S \tag{18}$$

where $\mathbf{z}[n] = \left[z_{-L_0}[n], \dots, z_{L_0}[n]\right]^T$, $l \in [-L_0, \dots 0, \dots L_0]$, and $z_l[n]$ is the inverse discrete time Fourier transform (DTFT) of $Z_l(f)$. $\mathbf{A}_S^\dagger$ is the generalized inverse matrix of $\mathbf{A}_S$. Thus, the corresponding relation between source signals and received signals is established. The separated signal can be recovered in an order manner from $\mathbf{y}[n]$.

In Section 3.1, the support set was roughly estimated by the tracking of sub-channels. The accurate estimation of the support set by dynamic programming is shown in Figure 3. The situation of sub-bands (-1~-L) is symmetrical with that of sub-bands (L~1), not shown in Figure 3.

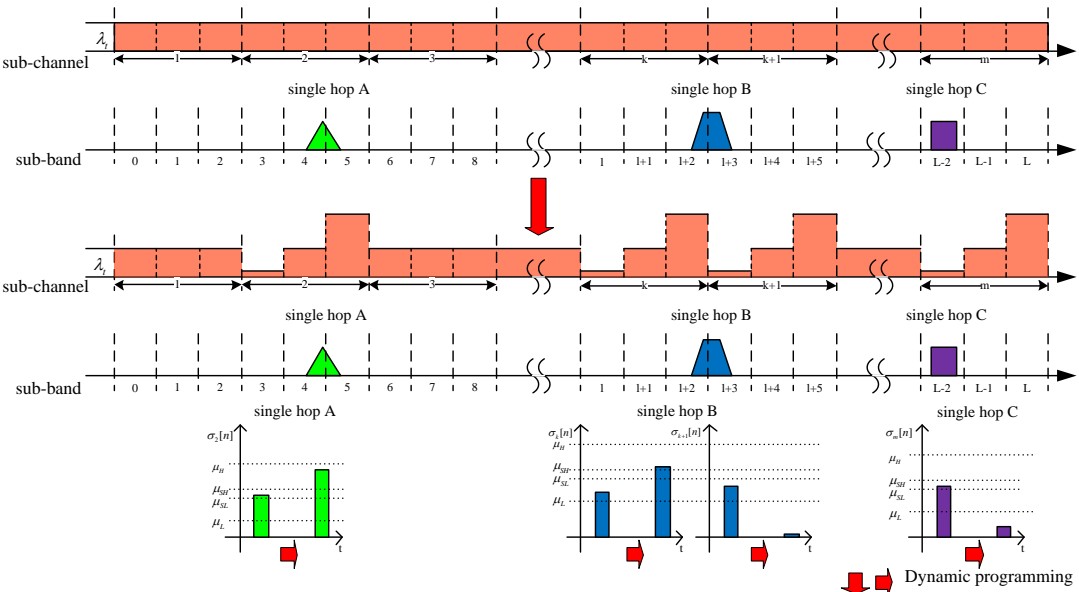

**Figure 3.** Dynamic programming strategy for support set estimation.

Without loss of generality, three sub-bands correspond to one sub-channel in Figure 3. In the tracking state, the amplitudes of multi-frequency functions $\lambda_{il}$ are equal for the sub-bands. Once the power information of the single hop is detected in the $i$th sub-channel by $\sigma_i[n]$, the amplitudes of multi-frequency functions for the $i$th sub-channel are dynamically programmed and other sub-channels stay the same.

Suppose that the corresponding sub-bands for the $i$th sub-channel are $k$, $k+1$ and $k+2$. In the tracking state, the amplitudes of multi-frequency functions are

$$\lambda_{i,k} = \lambda_{i,k+1} = \lambda_{i,k+2} = \lambda_t \tag{19}$$

The power estimation of the $i$th sub-channel is $\sigma_t$.

If the observed single hop does not cross sub-channels, there are 5 kinds of relationships between the single hop and sub-bands:

$H0$: The hop is only in the $k$th sub-band (as the single hop C in Figure 3).

$H1$: The hop crosses the $k$th sub-band and $(k+1)$th sub-band.

$H2$: The hop is only in the $(k+1)$th sub-band.

$H3$: The hop is crosses the $(k+1)$th sub-band and $(k+2)$th sub-band (as the single hop A in Figure 3).

$H4$: The hop is only in the $(k+2)$th sub-band.

In the proposed system, the relationships are identified by dynamic programming.

First, a gain factor $\beta(\beta > 1)$ is applied for the amplitudes of multi-frequency functions. Thus we obtain

$$
\begin{cases}
\lambda_{i,k} = \frac{1}{\sqrt{\beta}}\lambda_t \\
\lambda_{i,k+1} = \lambda_t \\
\lambda_{i,k+2} = \sqrt{\beta}\lambda_t
\end{cases}
\tag{20}
$$

For the cross-sub-band single hop, the power in each sub-band should not be less than $\gamma(\gamma < 1)$ of the total power. Generally, $\gamma$ is not less than 5%. The four thresholds in Figure 3 for identification can be expressed as

$$
\begin{cases}
\mu_L = \frac{1-\gamma}{\beta}\sigma_t \\
\mu_{SL} = (1-\gamma)\sigma_t + \frac{\gamma}{\beta}\sigma_t \\
\mu_{SH} = (1-\gamma)\sigma_t + \gamma\beta\sigma_t \\
\mu_H = (1-\gamma)\beta\sigma_t
\end{cases}
\tag{21}
$$

Suppose the power estimation is $\sigma_d$ after dynamic programming. The identification for the relationships can be expressed as

$$
\begin{cases}
H0: & \sigma_d < \mu_L \\
H1: & \mu_L \leq \sigma_d < \mu_{SL} \\
H2: & \mu_{SL} \leq \sigma_d < \mu_{SH} \\
H3: & \mu_{SH} \leq \sigma_d < \mu_H \\
H4: & \sigma_d \geq \mu_H
\end{cases}
\tag{22}
$$

If the observed single hop crosses sub-channels, the situations are opposite between two adjacent sub-channels after dynamic programming (as single hop B in Figure 3), which is easy to distinguish.

The result of identification is the estimation of support sets. Based on Equation (18), the multiple signal can be separated in the order of sub-bands. The algorithm for the separation is shown in Figure 4.

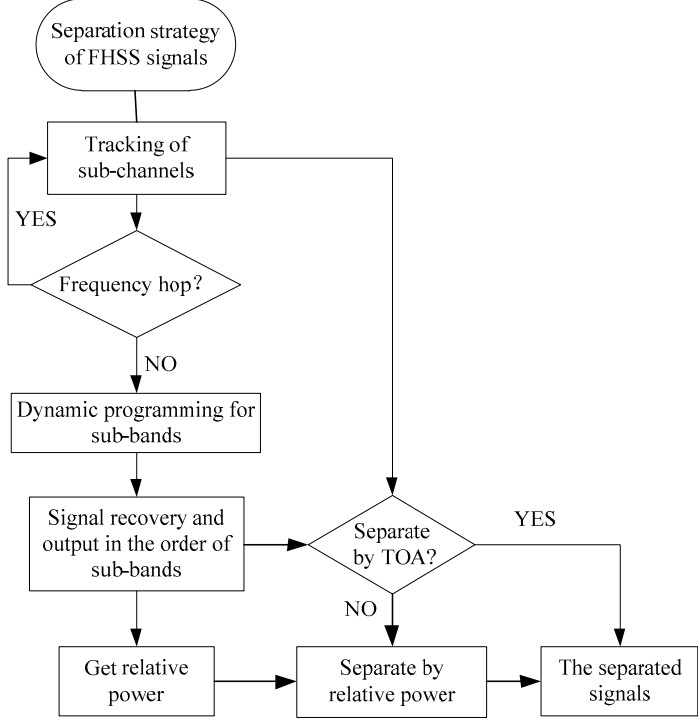

**Figure 4.** Separation strategy of frequency hopping (FH) signals.

The detail of the strategy can be summarized as follows:

**Input:** Received signals.

**Step 1:** Detect the sub-channels of FH signals by $\sigma_i[n]$ based on Equation (12).

**Step 2:** If a frequency hop occurred, obtain the TOA estimation of the new hop by Equation (13).

**Step 3:** Refresh the amplitudes of multi-frequency functions of the sub-bands by dynamic programming in the sub-channels that were detected in Step 1. Extract the sub-bands as the support set by Equation (22).

**Step 4:** Recover the signal in the order of sub-bands by Equation (18).

**Step 5:** Get the relative power of $z_i[n]$ in Step 4.

**Step 6:** Separate the multiple signals by TOA, obtained in Step 2. If the signals cannot be separated by TOA, take relative power into consideration for the separation.

**Output:** Separated signals $z_r[n], r = 1, \ldots, R$.

## 4. Simulations and Discussion

In this section, signal tracking and sorting performance are analyzed by numerical simulations. The intermediate result of TOA estimation and dynamic programming are also discussed. The simulation settings are shown in Table 1.

**Table 1.** Simulation settings.

| Parameter | Value |
| --- | --- |
| Sub-band number $L$ | 195 |
| Periodic sequence frequency $f_p$ | 51.3 MHz |
| Baseband sampling rate $f_s$ | 51.3 MHz |
| Sub-channel number $m$ | 32, 50 |
| Amplitude of smoothing filter | 1 |
| Carrier frequency | [2, 7 GHz] |
| Hop rate | 10 khop/s |
| Single hop bandwidth $B$ | 30 MHz |
| Mode type | DQPSK |
| Shaping filter | Raised cosine FIR |

Without loss of generality, a differential quadrature reference phase shift keying (DQPSK) modulated multiple signal is applied in this section. The carrier frequency of source signals is transformed to [0, 5 GHz] as the input of our system and the sub-band number is $L$ = 195. Thus, the Nyquist sampling rate is $f_{nyq} = 10$ GHz. The periodic frequency is $f_p = f_{nyq}/L \approx 51.3$ MHz. The baseband sampling rate is $f_s = f_p = 51.3$ MHz, which is the theoretical minimum. The total sampling rates of our system are $f_s \times 32 = 1.6416$ GHz for $m$ = 32 and $f_s \times 50 = 2.565$ GHz for $m$ = 50, which is much lower than the Nyquist sampling rate. The number of sampling points for a single hop is $f_s/1 \times 10^4 = 5130$, which is enough for further processing. Each signal is shaped by the raised cosine finite impulse filter (FIR), which has been widely used in practice.

### 4.1. FH Signal Tracking

In this section, the FH signal tracking and dynamic programming performance is analyzed. The simulation results of signal tracking are shown in Figure 5. In the present study, the number of sub-channels is set as $m$ = 50 and each sub-channel contains two pairs of sub-bands for signal tracking. The SNR for each signal is 10 dB on the equivalent narrow band.

Four hops of each source are shown in Figure 5 as examples. It can be seen that the power estimation of sub-channels is effective. Hopping between sub-channels is taken for the TOA estimation, marked by a red line in Figure 6. During a single hop, the power estimation is not stable, because the length of the smoothing filter in the time domain is limited and the noise of all the sub-bands is superimposed.

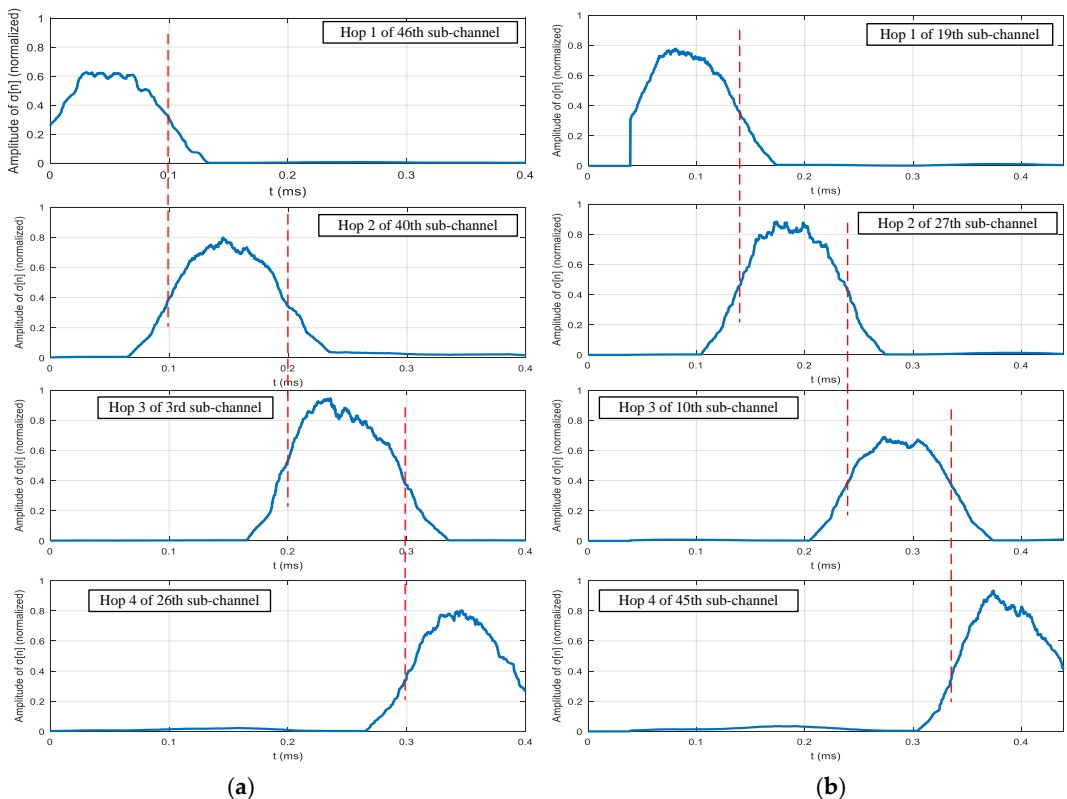

**Figure 5.** Results of signals tracking: (**a**) source 1; (**b**) source 2.

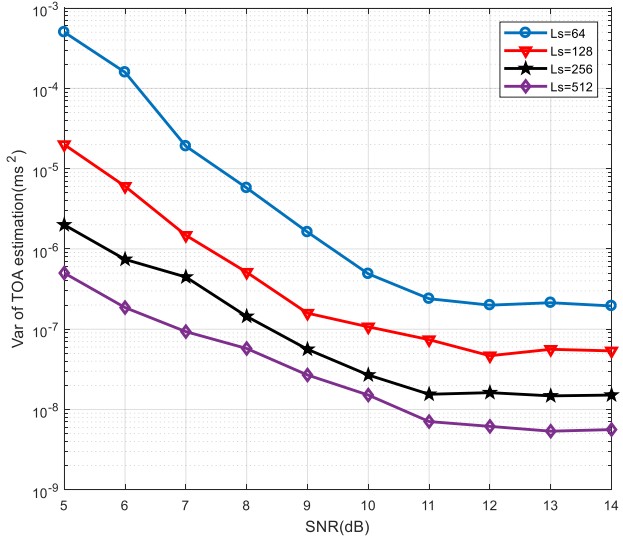

**Figure 6.** Performance of the TOA estimation.

The performance of the TOA estimation is shown in Figure 6, where *Ls* represents the length of the smoothing filter. It can be concluded that SNR is the main factor affecting the TOA estimation when it is lower than 10 dB, because the disturbance of noise cannot be removed by the smoothing filter. The TOA estimation is relatively accurate and stable when SNR is higher than 10 dB, and the main factor that affects the estimation is the length of the smoothing filter. The longer the smoothing filter, the better the estimation, because it represents the accumulation of time in the time domain. Based on Equation (12), long-term estimation is closer to real TOA, but it loses real-time performance.

In the application, a compromise between accuracy and real-time requirement should be made for a better performance and the filter delay should be eliminated.

Figure 7 shows the support set estimation by dynamic programming, where $P_d$ is the success rate of the estimation. The length of the smoothing filter for this simulation is $Ls = 256$. The number of sub-channel is set as $m = 32$ and each sub-channel contains three pairs of sub-bands.

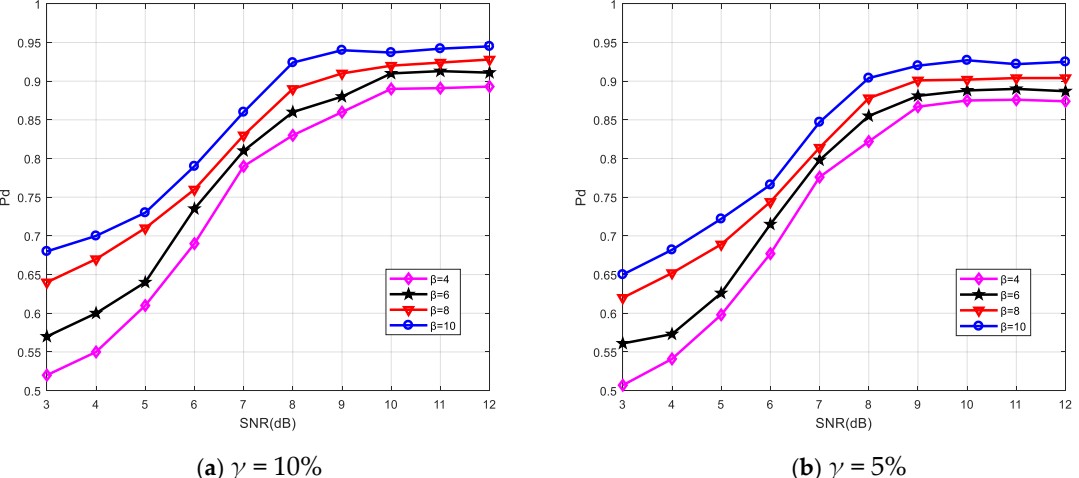

(**a**) $\gamma = 10\%$                                    (**b**) $\gamma = 5\%$

**Figure 7.** Performance of the support set estimation by dynamic programming.

It can be concluded that with increased SNR, the success rate becomes higher. The gain factor has a significant impact on the probability of estimation. When the SNR is higher than 9dB and the gain rate is higher than 8, the success rate is stable at over 90%. As the power of a single hop is relatively stable and the MWC system spreads all the information of input signals, including noise, to the baseband, the inherent characteristics of the noise limit the sensitivity of estimation. Comparing Figure 7a,b, it can be seen that the signal power limit ratio of the sub-bands $\gamma$ is not the main factor for the estimation. The lower the ratio, the worse the success rate. For practical applications, the choice of gain factor should consider the channel condition and the requirement of estimation accuracy.

*4.2. FH Signal Sorting*

In this section, the performance of FH signal sorting is analyzed. The performance of separation is evaluated in terms of the average signal-to-interference ratio (SIR).

$$SIR = 10 \log_{10} \left( \frac{\sum\limits_{n=1}^{N} E\{s_n^2(t)\}}{\sum\limits_{n=1}^{N} E\{(s_n(t) - \hat{s}_n(t))^2\}} \right) \tag{23}$$

where $s_n(t)$ is the source signal, and $\hat{s}_n(t)$ is the separated signal, which is an estimation of $s_n(t)$. The higher the SIR, the more similar the separated signal to the source, which means better performance.

Figure 8 shows the performance of FH signal sorting. In the present study, the number of sources is three. For our system, the number of sub-channels is set as $m = 50$ and each sub-channel contains two pairs of sub-bands for signal tracking, and $Ls = 256$, $\beta = 8$, $\gamma = 10\%$. Compared with the subspace-based algorithm [3], the proposed algorithm performs better. As the subspace-based algorithm is based on calculating a mixing matrix and comparative power, the error of matrix estimation will lead to amplitude fluctuation of the separated signals. The proposed algorithm can avoid the influence of sub-bands that do not contain signals and filter the noise out of the band. In Figure 8, the orthogonal matching pursuit (OMP) algorithm [31] for recovery is taken as a reference. The performance of OMP does not involve separation. It is shown here as a performance upper bound under ideal conditions.

The single hop is considered long enough as a stationary signal. The SIR performance of the proposed algorithm is a little worse than the OMP recovery method, because the noise of the sub-bands is amplified. The performance improves at high SNR and gets closer to recovery, because the noise is no longer the main cause of performance degradation. Our method has the advantages of fewer calculation requirements and adjustability.

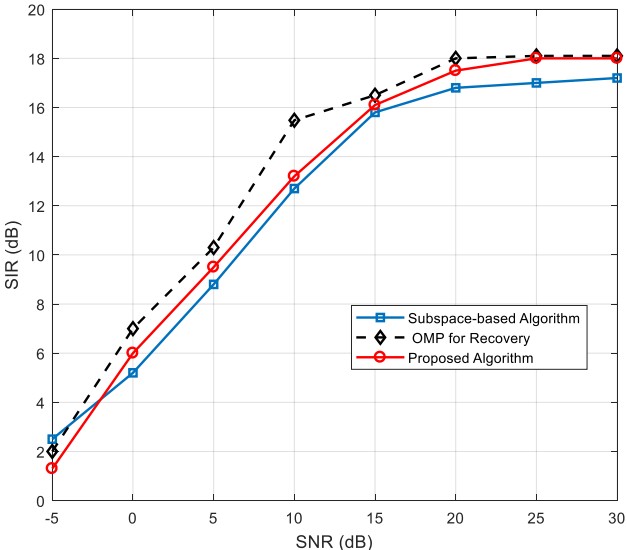

**Figure 8.** Performance comparison against SNR.

Figure 9 shows the performance of FH signal separation against number of sources $Ns$. In the present study, the number of sub-channels is set as $m = 50$ and each sub-channel contains two pairs of sub-bands for signal tracking; $Ls = 256$, $\beta = 8$ and $\gamma = 10\%$. It can be concluded that with more sources, the SIR for separated signals decreases, because the signals interfere with each other and the estimation of relative power degrades. Thus, the sparsity of signals should be considered in the application of CS systems.

Furthermore, FH code sequences always have the characteristic of a wide gap. The probability of two adjacent hops in different channels can be relatively high. It is beneficial for our system to sense changes in the hops.

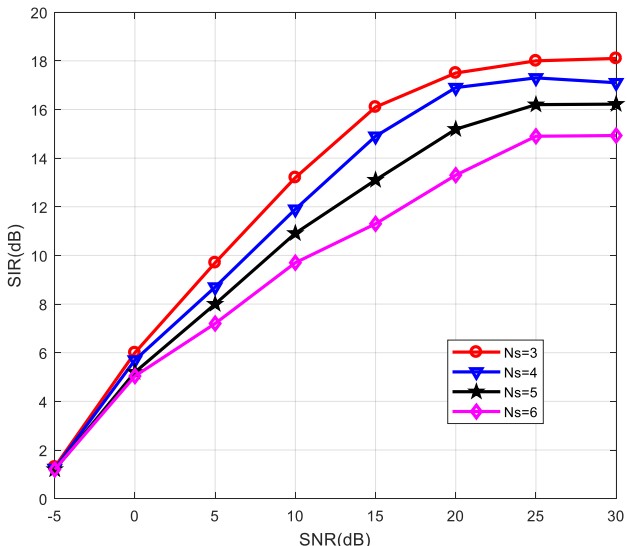

**Figure 9.** Performance of separation against number of source.

## 5. Conclusions

In this paper, we propose a dynamic programming MWC structure for non-cooperative tracking and sorting of FH signals. First, based on the power estimation in the time domain, the sub-bands that contain FH signals can be tracked in real time. The number of sampling points and amount of calculation are significantly reduced compared to the traditional methods. Second, we can separate multiple FH signals by the tracking results of TOA and comparative power. The theoretical analysis and numerical simulations demonstrate the validity and correctness of our method. The experimental results illustrate that CS has obvious advantages in blind tracking and sorting of signals. The SIR of the separation results is relatively high, which is effective for further processing. In future research, we would like to explore DOA estimation by MWC and other CS structures to establish a more comprehensive sorting system for multiple FH signals.

**Author Contributions:** Z.L. and P.Y. conceived the idea and wrote the paper. Z.L. designed and performed the experiments and analyzed the results. L.Z. and H.X. contributed analysis tools. L.Z. and H.D. reviewed the paper.

**Funding:** This research was funding by National Natural Science Foundation of China. Project Number: 61571452.

**Acknowledgments:** Our work has received important English editing from MDPI.

**Conflicts of Interest:** The authors declare no conflict of interest.

## Appendix A

This part shows the precise description of the mixing function $p_i(t)$.

The basic structure of MWC is shown in Figure A1 [31]. We assume that $x(t)$ is a continuous time signal with a range within $\mathcal{F} = [-1/2T, 1/2T)$, where the Nyquist frequency $f_{NYQ} = 1/T$.

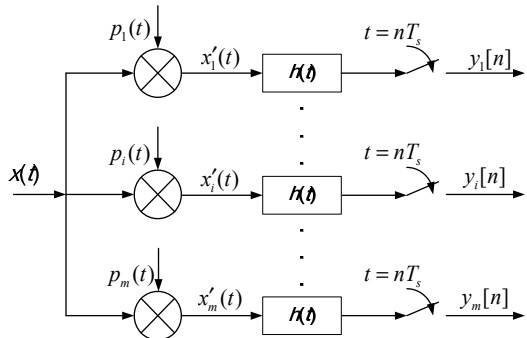

**Figure A1.** MWC under-sampling system.

MWC consists of $m$ parallel channels. In each channel, the multiband signal $x(t)$ is multiplied with a different mixing function, and $p_i(t), i = 1, 2, \ldots, m$. Each mixing function is periodic with the period $T_p = 1/f_p$. $p_i(t)$ is set as a sign function for each of the $M$ equal time intervals and other forms are possible, as the system only requires $p_i(t)$ periods, which is expressed as follows

$$\alpha_{ik} = p_i(t), \qquad k\frac{T_p}{M} \le t \le (k+1)\frac{T_p}{M} \tag{A1}$$

where $0 \le k \le M - 1$ and $\alpha_{ik} \in \{+1, -1\}$.

The purpose of this mixing function is to create aliases, so that the mixed signals $x'_i(t)$ have information about the entire spectrum in the baseband $\mathcal{F}_p = \left[-\frac{f_p}{2}, \frac{f_p}{2}\right]$. Taking the $i$th channel as an example, the Fourier expansion of $p_i(t)$ is as below

$$p_i(t) = \sum_{l=-\infty}^{+\infty} c_{il} e^{j2\pi l f_p t} \tag{A2}$$

where $c_{il}$ is the Fourier series coefficient and is expressed as

$$c_{il} = \frac{1}{T_p} \int_0^{T_p} p_i(t) e^{-j2\pi l f_p t} dt \tag{A3}$$

The Fourier transform of the mixing signal $x_i'(t) = x(t)p_i(t)$ is

$$x_i'(f) = \sum_{l=-\infty}^{+\infty} c_{il} X(f - l f_p) \tag{A4}$$

Suppose that the filter $H(f)$ is an ideal rectangular function. Consequently, the uniform sequence $y_i[n]$ has only frequencies in the $[-f_s/2, f_s/2]$. The DTFT of $y_i[n]$ is expressed as

$$Y_i\left(e^{j2\pi f T_s}\right) = \sum_{l=-L_0}^{+L_0} c_{il} X(f - l f_p), \qquad f \in \mathcal{F}_s \tag{A5}$$

where $L_0$ is selected as the smallest integer as

$$-\frac{f_s}{2} + (L_0 + 1) f_p \geq \frac{f_{nyq}}{2} \rightarrow L_0 = \left\lceil \frac{f_{nyq} + f_s}{2 f_p} \right\rceil - 1 \tag{A6}$$

where $f_s = 1/T_s$ is the sampling rate; and $\lceil \bullet \rceil$ means round up.

## Appendix B

This part shows a brief description of the expectation restricted isometry property (ExRIP).

Assume that $\delta_K$ is a isometry constant, $0 < \delta_K < 1$. The definition of restricted isometry property (RIP) is [37]

$$(1 - \delta_K) \|\mathbf{u}\|^2 \leq \|\Phi \mathbf{u}\|^2 \leq (1 + \delta_K) \|\mathbf{u}\|^2 \tag{A7}$$

where $\delta_K$ is the minimum value satisfying formula. If Equation (A7) holds for any *K*-sparse vector $\|\mathbf{u}\|$, then the matrix $\Phi$ satisfies the RIP of isometry constant $\delta_K$. Definition [37]: A matrix has the ExRIP, if Equation (A7) holds with probability at least p for K-sparse random vectors $\mathbf{u}$ whose support is uniformly distributed and whose non-zeros are independent identically distributed random variables.

The detailed procedure to prove that MWC has ExRIP has been proposed in [37].

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
