# Peer review of "Frequency Hopping Signals Tracking and Sorting Based on Dynamic Programming Modulated Wideband Converters"

_applsci, doi:10.3390/app9142906_

Round 1

Reviewer 1 Report

This manuscript tackles a quite interesting problem and sounds correct from a technical and mathematical viewpoint. The contribution is limited to the proposal of a method, whose efficiency is shown through few numerical results. 

The contribution is not clear to me in many aspects. The introduction is vague, as many concepts are not well defined, first of all the concept of Frequency Hopping. Therefore, the paper is not self-consistent. The manuscript is based on the assumption that compressed sensing (CS) is not suitable for time-varying signals. However, this is not true, in the sense that a widespread literature on dynamic CS has arisen in the last year. The proposed algorithm is tested through numerical simulations, in which however the proposed method is not compared to other methods, expect for Figure 8, where it is shown to be not better than other methods. Therefore, the real efficiency of the method is not clear. Moreover, the language is often not correct.

Based on these observations, the Authors should make this work more solid by deeply revising the presentation of concepts and results.

In the following, I highlight some errors and unclear points.

P.1

- The introduction is often vague, because some main concepts are not defined and the references are not illustrated. Some examples:

What is "Frequency Hopping"? Is the described and illustrated, but not defined. Definition is fundemental to avoid misleading. 

"The tracking of FH signals can be regarded as the real-time estimation of hopping time and frequency." This is misleading: the paper stars talking about FH. What is hopping time? And then, what the Authors aim to track?

What is the "network sorting"? The statement: "The sorting is a problem of information processing based

on the characteristics and independence of signals." is too vague. 

The methods in [1-7] should be briefly illustrated. "However, the methods in [1-7] cannot

process the signal in real time": why? Sparsity and compressed sensing (CS) are not defined and references are missing. The Authors should review the introduction by assuming that the reader might not be familiar with these concepts. The paper should be autonconsistent, so that even a reader not familiar with these concepts can follow the main points.

- "the estimation of parameter" should be "the estimation of parameters"?

P.2

- "The compressed sensing (CS)" should be "Compressed sensing (CS)"

- CS should be defined and/or associated with a bibliographic reference when introduced.

- Regarding CS: "However,these methods cannot process the signals in real time". I do not agree, in the sense that in the last years there has been a widespread reserach on dynamic CS, which is focused exaclty on tracking sparse signals, see the paper "Online optimization in dynamic environments: a regret analysis for sparse problems" (CDC 2018) and the reference therein. Therefore, CS might be applied to the proposed problem. The Authors should mentiong dynamic CS as possibility to tackle the problem.

- "The sparsity of FH signals changes rapidly, which is not suitable for traditional methods": it is more correct to say that traditional methods are not suitable for FH. 

- "2. The structure of sensing matrix is always fixed" should be " The structure of the sensing matrix". Moreover, this sentence is vague. What does it mean that it is always fixed?

- "4. The requirements for caching data of CS has critical real-time requirement": what does this mean? where this point has been studied? This statement is vague.

- "... [26], A method"

P.3

- Why "compressive sampling" is now used instead of the acronym CS already defined (provided that compressed sensing= compressive sampling)? This is misleading

- "The contributions of this work are as follows:" the 3 points that follow are more conclusions than contributions. As contributions I would expect to know how these conclusions are obtained (theoretical proofs? numerical simulations? real experiments?)

- "Assuming that there are R FH signals impinging onto the single antenna." Grammar is wrong.

P.4

- Section 3.1: the tracking phase is not clear to me, as the Authors have not defined rigorously what they want to trakc. I guess they want estimate online x(t), but the proposed scheme deos acquire, mixes, and samples x(t), producing compressed measurements y(t). The proposed scheme for "tracking of the power change between sub-channels": is this the aim of tracking? This should be stated more clearly and before proposing the scheme.

- "In order to apply CS for the tracking of FH signals, the system should be able to cope with the changes in sparsity and update the processing results rapidly". The Authors aim to track time-varying signals using CS, and they say that this is not done in "classical" CS. This is true, but there is a widespread recent literature on dynamic CS and tracking of time-varying sparse signals. The Authors seem to ignore this literature.

- " To make the system more suitable and feasibility for ..."

- "To make the system more suitable and feasibility for FH signals, the dynamic programming MWC structure is proposed as follows:" what follows is a figure, therefore this is not the right way to refer to it. It should be "To make the system more suitable and feasibility for FH signals, a novel dynamic programming MWC is proposed, which is depicted in Figure 2."

- The function p(t) should be defined more precisely.

P.6

- " crad () denotes the number of set." This definition is obscure. I suppose that card is for the cardinality of the set.- 

- I do not understand the contribution of Section 3.1: it seems to review known concecpts and I do not get the essence of the tracking algorithm. (13) and (14) are not clear. "It is more efficient to process directly in time domain than that in frequency domain, which is conducive for signal tracking." this sentence is vague.

P.8

- "Supposing that the corresponding sub-bands for ith sub-channel are k, k+1 and k+2."

- "Supposing the power estimation is .. after dynamic programming.

P.12

- " In Figure 9, The " 

- OMP is in Figure 8 and not 9, to the best of my understanding.

Reviewer 2 Report

- The literature review on compressive sensing is not sufficient and more discussion and details on the sparse recovery algorithms are needed.

- On page 4, the notion of ExRIP needs more elaboration. Although the theory behind restricted isometry property has been around for almost two decades, the ExRIP is not that much well-known. So, I suggest the authors provide a short description of it.

- Page 7, the equations are not new. I suggest the authors cite some of Eldar's work here regarding the Xampling problem.

- Page 9, I suggest the authors refer to the related equations for the steps of their proposed strategy.

- Page 9, Section 4: Please provide more discussion and reasoning behind using the sampling rates of 1.6416 GHz and 2.565 GHz.

-  Page 10, Tab. 1: Some discussion on the baseband sampling rate of 51.3MHz is needed here.

- Page 7, Fig. 7, I suggest the authors show the False alarm rate in support recovery, as well. I suggest the authors use the same sort of figures as 

Shekaramiz, M., Moon, T.K. and Gunther, J.H., 2019. Bayesian Compressive Sensing of Sparse Signals with Unknown Clustering Patterns. Entropy21(3), p.247.

- Page 12, Fig. 8: The authors used the comparison with OMP. Is it the modified OMP :ReMBO algorithm? Please provide more detail on this along with the setting of the algorithm.

Minor editorial concerns:

- The format and font size of the equations should be changed.

- On page 6, Eq(13), I suggest the authors use Card instead of Crad, as Card is more representative of the cardinality of a set.

- Page 12, Eq. 22, It might be better to use "log" instead of "lg"

- Please proofread the references. Some lower case /upper case issues exist there with some minor typos. For example:

 Ref. 3

Ref 4.: DOA instead of doa

Ref. 6 and 8; IET instead of Iet

Ref. 9: Bayesian instead of bayesian

Ref. 12: ICA instead of ica

Ref. 17, 18, 20

Round 2

Reviewer 1 Report

The Authors have done a remarkable revision work, and  they addressed my concerns. I believe that the the manuscript is now easier to read and all the points that were not clear (like figures explanations) are now fixed. It seems to me that all the grammar errors and typos have been corrected. In my opinion, the present version of the manuscript is eligible for  publication in Applied Sciences.

Reviewer 2 Report

The authors have provided reasonable and satisfactory responses to my earlier comments.